# Determinants of intimate partner violence against women in Ethiopia: A multi-level analysis

Tenaw Yimer Tiruye[1,2]*, Melissa L. Harris[2], Catherine Chojenta[2], Elizabeth Holliday[3], Deborah Loxton[2]

1 Public Health Department, College of Health Sciences, Debre Markos University, Debre Markos, Ethiopia, 2 Research Centre for Generational Health and Ageing, School of Medicine and Public Health, Faculty of Health and Medicine, the University of Newcastle, Newcastle, Australia, 3 School of Medicine and Public Health, Faculty of Health and Medicine, University of Newcastle, Newcastle, Australia

* tenaw.yimer@uon.edu.au, tenyim09@gmail.com

**Data Availability Statement:** Data cannot be shared publicly because data was provided by a third party (the Ethiopian Demographic and Health Survey). DHS data files are available for research/ analysis, at no cost, with the condition that the user

## Abstract

Intimate partner violence (IPV) continues to be a major public health problem globally. Although Ethiopia has a high prevalence of IPV, previous studies in this country have only investigated individual-level determinants of IPV within small geographic areas. The current study aimed to identify the individual-, relationship-, community-, and societal-level determinants of IPV directed against women in Ethiopia since women are predominantly affected. A retrospective analysis of nationally representative data from the 2016 Ethiopian Demographic and Health Survey (EDHS) was conducted. A sample of 3,897 married women of reproductive age (15–49 years) who participated in the domestic violence module of the survey were included in the analysis. Three-level mixed-effects multilevel logistic regression models were used to estimate the individual-, relationship-, community-, and societal-level determinants of IPV. Variability at the community- and societal-level were also assessed. About 1,328 (34.1%) of 3,897 participants reported experiencing IPV (a composite measure of physical, sexual and emotional abuse). In adjusted models, the odds of lifetime IPV experience were higher among women who were older, were married before the age of 18 years, witnessed inter-parental violence during their childhood, had a partner who drank alcohol, and lived in a community with high IPV accepting norms. Alternatively, the odds of IPV were lower among women who had decision-making autonomy in the household, had the same or lower educational attainment as their partner, and lived in a community with low proportions of educated women. These findings reveal that although individual-level factors were significant determinants of IPV, higher level factors, including female education and IPV acceptance in the community, were also important influences on this major public health issue in Ethiopia. These findings suggest combined interventions at different levels may reduce IPV in this country.

provides an abstract or a description of any project that will be using the data. Researchers can request approval to access the data underlying this study for the DHS program through online application. The application and data access procedures are explained below. For questions or comments about accessing The DHS Program data, users can email to archive@dhsprogram.com and for general inquiries, they can contact info@dhsprogram.com. Researchers must apply for a download account before they can download datasets. To register for a download account, they can go to: http://dhsprogram.com/data/new-user-registration.cfm and fill the following information: • Email address, name, name of institution and institution type, country, and phone number. • Project title, co-authors in the project, and abstract or description of study (minimum of 300 words and maximum of 2500 words). • Region and country they wish to do the research/analysis. For this study, we have selected 'sub-Saharan Africa' and 'Ethiopia'. • The type of data set (survey, GPS, HIV or SPA data) they wish to access. For our analysis, we have selected 'survey' data set. Once the above information is completed, researchers can submit the data set request. After access to the data is approved, researchers can login and download the data. Researchers will come across the recoded survey data with different file names in different file formats. For our study, 'etir70dt' was used.

**Funding:** This study is partially funded by the University of Newcastle, which has provided a scholarship for the student researcher and supported him in obtaining statistical support and training. Dr. Melissa Harris is funded by an Australian Research Council Discovery Early Career Research Award.

**Competing interests:** The authors have declared that no competing interests exist.

## Introduction

Intimate partner violence (IPV) is a 'health hazard' [1] that continues to be a global public health problem with higher prevalences in low-income countries [1, 2]. Globally, one in every three women has experienced at least one form of IPV during her lifetime [3]. IPV has been found to be a universal problem across all contexts and countries, but the distribution of occurrence varies widely [2, 4]. A multicountry study showed that the prevalence of IPV was high in sub-Saharan Africa, where the magnitude reaches 66% of women (95% CI: 54–78) [5].

Ethiopia is one African country with a high prevalence of IPV, where the lifetime prevalence has been estimated at 20% to 78% in different areas [6]. In the face of high gender inequalities, IPV poses an increased burden on women's health in Ethiopia. Although the consequences are underexamined in this country, some existing evidence shows its potential for producing severe physical, emotional, and reproductive health problems [7–10]. Severe consequences of IPV include reduced maternal health care utilization and adverse child health outcomes [10–13]. As there are no IPV interventions in the country, comprehensive IPV intervention strategies are needed. This requires an understanding of the factors that are associated with IPV.

Previous research conducted on IPV in different countries shows that risk factors for IPV extend beyond the characteristics of the individuals involved [14]. According to the ecological framework, IPV occurs due to an interaction of factors at four levels: the individual, the relationship, the community, and the societal [14]. According to this framework, some individual-level factors alone such as a woman's education and autonomy may not be sufficient to protect against IPV unless these factors are communal and largely shared within the community where women live [15]. In addition, tolerant community norms regarding aspects of IPV, including acceptance of male superiority and perceptions of IPV as inevitable within a relationship, are basic factors that not only underlie the occurrence of IPV [16–19] but allow it to persist in society [5, 18] and reduce the effectiveness of intervention efforts [18]. Moreover, it is believed that societal-level determinants such as poverty, gender inequality, and political contexts not only affect the distribution of IPV but also moderate community-, relationship-, and individual-level risk factors [14, 18]. Their influence could be direct or indirect by affecting institutional systems, family decisions and gender roles [20].

The presence of hierarchical level causal factors necessitates the use of advanced analytical methods to accurately estimate the effect of IPV determinants in particular communities. This is principally helpful for countries like Ethiopia, which contains a population of over 80 ethnic groups living in diverse contexts [21]. However, no previous study of IPV in Ethiopia has investigated how factors operate at different levels as most previous studies [22–26] were focused mainly on individual characteristics. There is thus limited evidence regarding the effect of community- and societal-level determinants of IPV. Prior studies [22–26] in this country were also small in scale with inconsistent findings which lacked country-level representativeness for a large diverse community. Moreover, these studies either did not include some important variables, such as attitudes and norms around IPV, or did not use appropriate statistical models to reveal unbiased estimates. Therefore, the focus of this research was to examine the individual-, relationship-, community-, and societal-level determinants of IPV in Ethiopia using nationally representative secondary data and appropriate statistical models to account for the hierarchical data structure.

## Methods

### Data source

This study was based on data from the 2016 Ethiopian Demographic and Health Survey (EDHS), which was the year the domestic violence module was added. The EDHS was a

national survey conducted from January 18 to June 27, 2016. The EDHS data were collected using five questionnaires (household, women, men, biomarker and health facility). The collected data were recoded for easier access and analysis [27].

## Sample size and sampling procedures

The EDHS used two-stage stratified cluster sampling. In the first stage, 645 primary sampling units were sampled: 443 from rural areas and 202 from urban areas. In the second stage, on average 28 households from each primary sampling unit was selected using systematic random sampling. In total, 15,683 women aged 15–49 who reported ever being married (with a response rate of 95%) participated in the survey. For the domestic violence module, only one married woman per household was interviewed and 5,860 women (97% response rate) were interviewed [27]. The current study included women who reported ever being married and completed the IPV questionnaire (weighted sample = 3,897). The sampling weights used in the EDHS account for the complex sampling procedures (multi-stage stratified cluster sampling) that might cause an unequal probability of selection for certain areas or subgroups either due to design or coincidence. Hence, sampling weights were adjusted for differences in probability of selection and interview that allow extrapolation of results to the national level of representativeness [27].

## Measurement and variables

**Dependent variables.** IPV was measured using women's self-reported responses to questions based on the modified Conflict Tactic Scales of Straus [28]. Women were asked whether or not they had experienced the following acts within their relationship, perpetrated by their husband/partner for currently married women and recent husband/partner for previously married women. Those women who were married more than once were further asked about violence committed by any other husband/partner. Respondents were categorized as having experienced lifetime IPV if they reported experiencing at least one act of IPV since the age of 15 years [27]. Table 1 presents the questions used to assess IPV and the form of IPV the questions measuring.

**Independent variables.** This study was based on the concept of an ecological framework, which proposes that IPV occurs due to the interaction of factors at four levels: the individual,

**Table 1. The tool used to assess IPV in the 2016 Ethiopian Demographic and Health Survey.**

| Question/item | IPV type |
|---|---|
| Push you, shake you, or throw something at you? | Physical IPV |
| Slap you? | |
| Twist your arm or pull your hair? | |
| Punch you with his/her fist or with something that could hurt you? | |
| Kick you, drag you, or beat you up? | |
| Try to choke you or burn you on purpose? | |
| Threaten or attack you with a knife, gun, or any other weapon? | |
| Physically force you to have sexual intercourse with him even when you did not want to? | Sexual IPV |
| Physically force you to perform any other sexual acts you did not want to? | |
| Force you with threats or in any other way to perform sexual acts you did not want to? | |
| Say or do something to humiliate you in front of others? | Emotional IPV |
| Threaten to hurt or harm you or someone close to you? | |
| Insult you or make you feel bad about yourself? | |

the relationship, the community and the societal. Accordingly, potential determinants of IPV at each of the four levels were identified using previous similar research conducted globally [14, 15, 29–31] and in Ethiopia [24–26, 32, 33].

Level 1 variables: Individual-level variables considered in the analysis were age, age at first marriage, education, employment, religion, number of children, access to media, witness inter-parental violence, substance abuse, attitude to IPV, and wealth index. Relationship variables comprised women's decision-making autonomy (yes/no), who headed the household, educational difference and age difference between male and female partners. The individual and relationship level variables with their categories, measurement or definition are displayed in Table 2.

Level 2 variables: Community characteristics, which in this study is represented by individuals living in the same cluster, were included as level 2 variables. Place of residence was defined as urban or rural using original EDHS coding. Other variables were constructed by aggregating individual- or relationship-level characteristics. The aggregates for clusters were computed using mean (for normally distributed characteristics) and median (for variables that were not normally distributed) values for women in each category of a given variable. Finally, each community level variable was re-grouped into lower and higher categories (Table 2).

Level 3 variables: Two societal-level variables were also included in this analysis. These are the Multi-dimensional Poverty Index (MPI) and the Gender Empowerment Index (GEI). MPI is a measure of poverty that considers three dimensions of poverty (education, health and standard of living) and ten indicators with a given weight [34]. The data on the MPI for each of the 11 regions of Ethiopia were taken from the Oxford Poverty and Human Development Initiative [34] and regions were classified as low or high MPI based on deviation from the national average. The GEI is a composite measure of three dimensions and 15 indicators: women's attitude to IPV with 5 indicators, women's social independence with 7 indicators that include items related to women's education, media exposure, employment and ages at first birth and cohabitation, and women's decision-making autonomy with 3 indicators [35]. Methodological details on generating the three dimensions from 15 indicators can be obtained from the cited reference [35]. In the current study, the three dimensions were further reduced using the principal component analysis (PCA) to one continuous variable (GEI) and then classified as below or above the national average (Table 2).

## Data processing and analysis

Multilevel logistic regression models were used to estimate the effects of IPV determinants at the three specified levels, using a sample of 3,897 women nested in 639 communities nested in 11 regions. On average, each cluster/community had 6 women (range, 2 to 13). Multilevel analysis allows for the estimation of valid standard errors by adjusting for within-cluster correlation of the response variable [36]. Multilevel analysis also enables the estimation of community and regional variation in women's experience of IPV [37].

Four models were constructed. In Model I, the empty or unconditional model, no covariates were included. This model was used to estimate the random intercept at community and region level and the variation in the odds of IPV experience between communities and between regions. Then, subsequent models were constructed by adding covariates at each level on the preceding model, that is, in Model II. Individual- and relationship-level variables were included. Both individual- and relationship-level variables were considered as level one variables because in the EDHS only one woman per household was sampled [27] and hence household/relationship-level clustering may not exist. In Model III, community-level variables were

**Table 2. List of variables, their categories and definitions.**

| Level | Variable | Category/Measurement/Definition |
|---|---|---|
| Level 1: Individual- level variables | Age (years) | The age of the woman categorized as 15–19, 20–24, 25–29, 30–34, 35–39, 40–44, 45–49. |
| | Age at first marriage | Grouped as <18 years and ≥ 18 years. |
| | Educational status | Maximum educational level categorized as uneducated, primary, or secondary and above. |
| | Employment status | Current employment status of the woman classified as unemployed or employed. |
| | Religion | The religion that the respondent is following categorized as Christian, Muslim, or others. |
| | Witness to inter-parental violence | 'Yes' or 'no' based on their answer to the question, "As far as you know, did your father ever hit your mother?" |
| | Number of living children | Grouped as one or less, two to three, and ≥ four. |
| | Substance abuse | Classified 'yes' if respondent drinks alcohol, chews khat or smokes tobacco and 'no' otherwise. |
| | Partner drinks alcohol | Classified 'yes' if partner drinks alcohol and 'no' otherwise. |
| | Attitude on IPV | The attitude on IPV was measured based on the following five questions that men and women were asked about whether situations of hitting or beating a wife is justifiable: if she goes out without telling him; neglects their children; argues with him; refuses to have sex with him; and burns the food [27]. If they said 'yes' to any one of the above questions, they were categorized as having an unfavourable attitude and otherwise favourable attitude. |
| | Access to media | If respondent read a newspaper, listened to the radio, or watched television, they were categorized as have access and otherwise no access. |
| | Household wealth index | Measured based on the number and kind of goods households have and housing characteristics (drinking water, toilet facility, flooring material and availability of electricity) and was generated using principal component analysis (PCA) and classified into quintiles from 1 (very poor) to 5 (very rich) [27]. |
| | Age of partner | Categorised as under 25, 25–34, or ≥35. |
| Level | Variable | Category/Measurement/Definition |
| Level 1: relationship-level variables | Women's decision-making autonomy | Labelled 'yes' if she was involved in all decisions regarding her own health care, major household purchases and visits to her family or relatives [27]. |
| | Head of household | Based on the gender of the head of the household and classified as either woman or man. |
| | Educational difference | The educational status of the woman compared to her partner's educational status and classified as equal, lower or higher. |
| | Age difference | The age of the woman compared to her partner's age and classified as woman younger, same age, husband older by ≤5 years, or husband older by more than 5 years. |
| Level 2: community-level variables | Place of residence | Defined as urban or rural using original EDHS coding |
| | Early marriage | Categorized as high if the proportion of women married before 18 years of age was 60.0–100% and low if the proportion was 0–59.9% |
| | Female literacy | Categorized as low if the proportion of women who attended primary or secondary education was 0–36.4% and categorized as high if the proportion was 36.5–100% |
| | Community's level of acceptance towards IPV | Categorised as low if the proportion of women with an unfavourable attitude (having an IPV accepting attitude) in the community was 0–66.7% and categorized as high if the proportion was between 66.8% and 100% |
| | Women's decision-making autonomy | Categorized as low if the proportion of women's decision-making autonomy in the community was between 0–71.4% and high if the value ranged from 71.5 to 100% |
| Level 3: societal-level variables | Multi-dimensional Poverty Index (MPI) | Regions were classified as low or high MPI based on deviation from the national average |
| | Gender Empowerment Index (GEI) | Classified as below or above the national average |

added to Model II, and in Model IV, societal-level (region) variables were added to Model III. Model IV was the final model used to estimate measures of association.

The measures of association (fixed effects) were presented as odds ratios together with 95% CI. Statistical significance was declared using a p-value <0.05. In addition, the measure of variance (random effects), which is the measure of residual errors at individual level and community & regional variation, was reported in terms of the intra-class correlation coefficient (ICC) [36] and proportional change in variance (PCV) [38].

### Ethics statement

The original survey was conducted after being ethically approved by the National Research Ethics Review Committee (NRERC) of Ethiopia (*Ref. No*: *3.10/114/2016*). Prior to analysis, we obtained permission from the Demographic and Health Survey program and ethical approval from University of Newcastle Human Research Ethics Committee (*Ref. No*: *H-2018-0055*).

## Results

### General characteristics of respondents

In total, 3,897 (unweighted sample of 4,123) participants were included in the analysis. The majority of study participants were aged 25–29 years (23.2%), married before 18 years of age (62.2%), illiterate (61.5%), unemployed (50.1%), Christian (64.5%), married to a uneducated (47.2%) partner, and had the same educational level as their partner (62.1%). In total, 70.7% of participants had witnessed inter-parental violence during childhood and 67.3% had an IPV accepting attitude. About 69% of participants reported having no decision-making autonomy and 86% participants reported the husband was the head of the household. About 47.9% of individuals described themselves as having a habit of substance abuse and 62.2% had no access to media. Regarding community- and region-level characteristics, the majority of respondents were living in a community with rural residence (83.8%), high early marriage (52.1%), low female literacy (53.5%), low women's autonomy (54.9%), and high IPV accepting norms (52.0%) and in societies with high MPI (66.3%) and low GEI (86.6%) (Table 3). Table 4 shows IPV experience by different variables.

### Prevalence of different forms of IPV

Table 5 shows the estimated prevalence of different forms of IPV with 95% CI. The least prevalent form of IPV was sexual IPV (11.5%) and the most prevalent form was physical IPV (23.3%). About one in every three (34.1%) women had experienced at least one form of IPV in their lifetime.

### Determinants of IPV

Table 6 presents the results of the multilevel logistic regression analysis, which shows the measure of association (fixed effects) and the random intercepts for the experience of IPV. Model I (the empty or unconditional model) shows that there was a statistically significant variation in the odds of IPV experience between communities ($\sigma^2$ = 0.79, p-value <0.001) and between regions ($\sigma^2$ = 0.20, p-value <0.001). The ICC shows that IPV experience of women within the same community has a higher clustering (ICC = 23.1%) while low degree of clustering in the region (ICC = 4.6%).

In Model II, only individual- and relationship-level variables were added. The results showed that higher age, early age at first marriage, witnessing inter-parental violence during childhood, an IPV accepting attitude, higher educational attainment (compared to partner), and having a partner who drank alcohol were positively associated with IPV, while having decision-making autonomy was negatively associated with IPV. Adjusting for level one variables reduced the variance parameters; the PCV indicates that 25.1% and 5.0% of the variance in IPV experience across communities and across societies respectively was explained by the individual-level characteristics. The ICC in Model II indicated that after adjusting for individual and relationship factors, 19.2% and 4.7% of the variation in women's IPV experience was attributable to differences between communities and societies respectively.

**Table 3. Characteristics of study participants (n = 3,897).**

| Factor Group | Variable | Class | Weighted frequency | Percent |
|---|---|---|---|---|
| Respondent characteristics | Current age | 15–19 | 222 | 5.7 |
| | | 20–24 | 592 | 15.2 |
| | | 25–29 | 903 | 23.2 |
| | | 30–34 | 827 | 21.2 |
| | | 35–39 | 631 | 16.2 |
| | | 40–44 | 424 | 10.9 |
| | | 45–49 | 298 | 7.6 |
| | Age at first cohabitation | <18 years | 2424 | 62.2 |
| | | ≥18 years | 1473 | 37.8 |
| | Educational status | No education | 2397 | 61.5 |
| | | Primary | 1067 | 27.4 |
| | | Secondary+ | 433 | 11.1 |
| | Employment status | Not employed | 1952 | 50.1 |
| | | Employed | 1945 | 49.9 |
| | Religion | Christian | 2512 | 64.5 |
| | | Muslim | 1313 | 33.7 |
| | | Other | 73 | 1.9 |
| | Witness inter-parental violence | No | 2755 | 70.7 |
| | | Yes | 1142 | 29.3 |
| | Number of living children | One or less | 920 | 23.6 |
| | | 2–3 | 1123 | 28.8 |
| | | ≥4 | 1855 | 47.6 |
| | Substance abuse | No | 2031 | 52.1 |
| | | Yes | 1866 | 47.9 |
| | Wife beating attitude | No | 1273 | 32.7 |
| | | Yes | 2624 | 67.3 |
| Partner characteristics | Age of partner | Below 25 | 200 | 5.1 |
| | | 25–34 | 1298 | 33.3 |
| | | ≥35 | 2398 | 61.6 |
| | Partner's educational status | No education | 1840 | 47.2 |
| | | Primary | 1397 | 35.9 |
| | | Secondary+ | 660 | 17.0 |
| | Partner drinks alcohol | No | 2750 | 70.6 |
| | | Yes | 1147 | 29.4 |
| Household characteristics | Access to media | No | 2424 | 62.2 |
| | | Yes | 1473 | 37.8 |
| | Wealth index | Poorest | 748 | 19.2 |
| | | Poorer | 792 | 20.3 |
| | | Middle | 799 | 20.5 |
| | | Richer | 733 | 18.8 |
| | | Richest | 824 | 21.2 |
| Relationship level variables | Decision-making autonomy | No | 1203 | 30.9 |
| | | Yes | 2694 | 69.1 |
| | Head of the household | Male | 3357 | 86.2 |
| | | Female | 540 | 13.9 |
| | Educational difference | Women higher | 379 | 9.7 |
| | | Same | 2421 | 62.1 |

(*Continued*)

**Table 3.** (Continued)

| Factor Group | Variable | Class | Weighted frequency | Percent |
|---|---|---|---|---|
| | | Husband higher | 1097 | 28.1 |
| | Age difference | Women younger | 121 | 3.1 |
| | | Same age | 83 | 2.1 |
| | | Husband older by ≤5years | 1555 | 39.9 |
| | | Husband older by >5years | 2137 | 54.9 |
| Community/cluster level variables (n = 639) | Place of residence | Urban | 200 | 31.3 |
| | | Rural | 439 | 68.7 |
| | Early marriage | Low | 344 | 53.8 |
| | | High | 295 | 46.2 |
| | Female literacy | Low | 289 | 45.2 |
| | | High | 350 | 54.8 |
| | Women decision-making | Low | 317 | 49.6 |
| | | High | 322 | 50.4 |
| | IPV acceptability | Low | 377 | 59.0 |
| | | High | 262 | 41.0 |
| Societal/regional-level variables (n = 11) | MPI | Below national average | 7 | 63.6 |
| | | Above national average | 4 | 36.4 |
| | GEI | Below average | 5 | 45.4 |
| | | Above average | 6 | 54.6 |

Abbreviations: MPI = Multi-dimensional Poverty Index; GEI = Gender Empowerment Index

In Model III, after community level variables were added to Model II, the findings in Model II largely persisted, except IPV accepting attitude which lost its significant association with IPV. The result also revealed that three community level variables were found to have significant association with IPV–community level female literacy, community IPV accepting norm, and women decision-making autonomy in the community. The PCV in Model III implied that 63.3% of the variation in IPV experience between communities was explained by individual and community level characteristics. Likewise, 15.0% of the variation in IPV experience between societies was explained by individual and community level characteristics.

In Model IV, the final model, societal-level variables were added to Model III. After controlling for factors at all levels, women's age was significantly associated with IPV. Compared to women aged 15–19 years, women in higher age groups were more likely to report experiencing IPV. Effect sizes increased for higher ages. For example, compared to women aged 15–19 years, women aged 20–24 and 45–49 were about two (AOR 2.02, 95% CI: 1.35–3.07) and three times (AOR 3.31, 95% CI: 2.03–5.40) more likely to report experiencing IPV. Alternatively, being younger at first cohabitation was associated with an increased risk of IPV: women aged < 18 years at first cohabitation had 28% higher (AOR 1.28, 95% CI: 1.08–1.52) odds of IPV compared to women aged ≥18 years at first cohabitation.

Women who had witnessed inter-parental violence were about three and half times more likely (AOR 3.33, 95% CI: 2.80–3.96) to report IPV compared to women who had not witnessed inter-parental violence. Women with decision-making autonomy in the household were 19% less likely (AOR 0.81, 95% CI: 0.68–0.97) to report experience of IPV compared to women who had no decision-making autonomy. Regarding partner's behaviour, women who had a partner who drank alcohol were three times (AOR 3.00, 95% CI: 2.42–3.67) more likely to report experience of IPV compared to women who had a partner who did not drink alcohol.

**Table 4. IPV experience by different variables.**

| Factor Group | Variable | Class | IPV (n = 3,897) | | P-Value* |
| --- | --- | --- | --- | --- | --- |
| | | | **No** | **Yes** | |
| | | | **No (%)** | **No (%)** | |
| Respondent characteristics | Current age | 15–19 | 168 (6.5) | 54 (4.1) | 0.238 |
| | | 20–24 | 402 (15.6) | 191 (14.4) | |
| | | 25–29 | 616 (24.0) | 287 (21.6) | |
| | | 30–34 | 527 (20.5) | 300 (22.6) | |
| | | 35–39 | 402 (15.7) | 228 (17.2) | |
| | | 40–44 | 277 (10.8) | 147 (11.1) | |
| | | 45–49 | 178 (6.9) | 120 (9.1) | |
| | Age at first cohabitation | <18 years | 1016 (39.6) | 457 (34.4) | 0.045 |
| | | ≥18 years | 1553 (60.4) | 871 (65.6) | |
| | Educational status | No education | 1524 (59.3) | 873 (65.8) | 0.001 |
| | | Primary | 705 (27.5) | 362 (27.2) | |
| | | Secondary+ | 340 (13.2) | 93 (7.0) | |
| | Employment status | Not employed | 1319 (51.3) | 633 (47.7) | 0.196 |
| | | Employed | 1250 (48.7) | 695 (52.3) | |
| | Religion | Christian | 1647 (64.1) | 865 (65.2) | 0.011 |
| | | Muslim | 893 (34.7) | 420 (31.6) | |
| | | Other | 30 (1.2) | 43 (3.2) | |
| | Witness inter-parental violence | No | 2028 (78.9) | 727 (54.8) | <0.001 |
| | | Yes | 542 (21.1) | 601 (45.2) | |
| | Number of living children | One or less | 654 (25.5) | 266 (20.0) | 0.035 |
| | | 2–3 | 725 (28.2) | 398 (30.0) | |
| | | ≥4 | 1191 (46.3) | 664 (50.0) | |
| | Substance abuse | No | 1413 (55.0) | 618 (46.5) | 0.005 |
| | | Yes | 1156 (45.0) | 710 (53.5) | |
| | Wife beating attitude | No | 910 (35.4) | 363 (27.3) | 0.001 |
| | | Yes | 1659 (64.6) | 965 (72.7) | |
| Partner characteristics | Age of partner | Below 25 | 148 (5.8) | 52 (3.9) | 0.283 |
| | | 25–34 | 848 (33.0) | 450 (33.9) | |
| | | ≥35 | 1573 (61.2) | 826 (62.2) | |
| | Partner's educational status | No education | 1140 (44.4) | 700 (52.7) | <0.001 |
| | | Primary | 908 (35.3) | 489 (36.8) | |
| | | Secondary+ | 521 (20.3) | 139 (10.5) | |
| | Partner drinks alcohol | No | 1951 (75.9) | 799 (60.2) | <0.001 |
| | | Yes | 618 (24.1) | 529 (39.8) | |
| Factor Group | Variable | Class | IPV (n = 3,897) | P-Value* | |
| | | | No | | Yes |
| | | | No (%) | | No (%) |
| Household characteristics | Access to media | No | 1531 (59.6) | 893 (67.2) | 0.007 |
| | | Yes | 1038 (40.4) | 435 (32.8) | |
| | Wealth index | Poorest | 465 (18.1) | 283 (21.3) | <0.001 |
| | | Poorer | 504 (19.6) | 289 (21.7) | |
| | | Middle | 484 (18.8) | 315 (23.7) | |
| | | Richer | 491 (19.1) | 242 (18.2) | |
| | | Richest | 625 (24.3) | 199 (15.0) | |

(*Continued*)

**Table 4.** (Continued)

| Factor Group | Variable | Class | IPV (n = 3,897) | | P-Value* |
|---|---|---|---|---|---|
| | | | No | Yes | |
| | | | No (%) | No (%) | |
| Relationship level variables | Decision-making autonomy | No | 739 (28.8) | 464 (34.9) | 0.024 |
| | | Yes | 1830 (71.2) | 864 (65.1) | |
| | Head of the household | Male | 2200 (85.6) | 1157 (87.1) | 0.428 |
| | | Female | 369 (14.4) | 171 (12.9) | |
| | Educational difference | Women higher | 208 (8.1) | 171 (12.9) | 0.010 |
| | | Same | 1631 (63.5) | 790 (59.5) | |
| | | Husband higher | 730 (28.4) | 366 (27.6) | |
| | Age difference | Women younger | 84 (3.3) | 37 (2.8) | 0.624 |
| | | Same age | 47(1.8) | 36 (2.7) | |
| | | Husband older by ≤5years | 1030 (40.1) | 526 (39.6) | |
| | | Husband older by >5 years | 1409 (54.8) | 729 (54.9) | |
| Community/cluster level variables (n = 639) | Place of residence | Urban | 125 (36.2) | 75 (25.5) | 0.004 |
| | | Rural | 220 (63.8) | 219 (74.5) | |
| | Early marriage | Low | 194 (56.2) | 150 (51.0) | 0.188 |
| | | High | 151 (43.8) | 144 (49.0) | |
| | Female literacy | Low | 159 (46.1) | 130 (44.2) | 0.636 |
| | | High | 186 (53.9) | 164 (55.8) | |
| | Women's autonomy | Low | 155 (44.9) | 162 (55.1) | 0.010 |
| | | High | 190 (55.1) | 132 (44.9) | |
| | IPV acceptability | Low | 223 (64.6) | 154 (52.4) | 0.002 |
| | | High | 122 (35.4) | 140 (47.6) | |
| Societal/regional-level variables (n = 11) | MPI | Below national average | 4 (66.7) | 3 (60.0) | 0.652[¥] |
| | | Above national average | 2 (33.3) | 2 (40.0) | |
| | GEI | Below average | 2 (33.3) | 3 (60.0) | 0.392[¥] |
| | | Above average | 4 (66.7) | 2 (40.0) | |

*P-value was based on chi-squared test; IPV = Intimate Partner Violence; MPI = Multi-dimensional Poverty Index

[¥]P-value was based on Fisher's exact test; GEI = Gender Empowerment Index

Regarding educational differences between spouses, women with the same educational attainment as their partner were 55% (AOR 0.45, 95% CI: 0.26–0.79) and women with a lower educational level than their partner were 69% (AOR 0.31, 95% CI: 0.11, 0.89) less likely to have experienced IPV compared to women who had a higher educational attainment than their partner.

**Table 5. Prevalence of different forms of IPV against women.**

| Form of IPV | Weighted prevalence | 95% CI |
|---|---|---|
| Physical IPV | 23.3% | (21.1%, 25.6%) |
| Sexual IPV | 11.2% | (9.4%, 13.1%) |
| Emotional IPV | 22.7% | (20.2%, 25.2%) |
| Physical, sexual or emotional IPV | 34.1% | (31.3%, 36.8%) |

Abbreviations: IPV = Intimate Partner Violence; CI = Confidence Interval

**Table 6. Multilevel logistic regression analysis of individual-, relationship-, community- and societal-level factors associated with IPV.**

| Group | Variable | Class | Model I | Model II AOR (95% CI) | Model III AOR (95% CI) | Model IV AOR (95% CI) |
|---|---|---|---|---|---|---|
| Level-I variables | Current age | 15–19 | | 1 | 1 | 1 |
| | | 20–24 | | 2.07 (1.37, 3.13) | 2.02 (1.34, 3.06) | 2.02 (1.35, 3.07)** |
| | | 25–29 | | 2.08 (1.39, 3.14) | 2.05 (1.36, 3.08) | 2.05 (1.37, 3.09)** |
| | | 30–34 | | 2.52 (1.66, 3.84) | 2.45 (1.61, 3.73) | 2.46 (1.62, 3.73)*** |
| | | 35–39 | | 3.14 (2.05, 4.82) | 3.04 (2.00, 4.68) | 3.05 (2.00, 4.68)*** |
| | | 40–44 | | 2.54 (1.59, 4.03) | 2.43 (1.53, 3.86) | 2.42 (1.52, 3.85)*** |
| | | 45–49 | | 3.41 (2.09, 5.56) | 3.32 (2.03, 5.41) | 3.31 (2.03, 5.40)*** |
| | Age at first cohabitation | <18 years | | 1.29 (1.08, 1.52) | 1.28 (1.07, 1.53) | 1.28 (1.08, 1.52)* |
| | | ≥18 years | | 1 | 1 | 1 |
| | Educational status | No education | | 1 | 1 | 1 |
| | | Primary | | 0.69 (0.42, 1.14) | 0.66 (0.40, 1.08) | 0.66 (0.40, 1.10) |
| | | Secondary+ | | 0.43 (0.17, 1.11) | 0.42 (0.16, 1.06) | 0.42 (0.17, 1.08) |
| | Witness inter-parental violence | No | | 1 | 1 | 1 |
| | | Yes | | 3.37 (2.83, 4.01) | 3.34 (2.81, 4.00) | 3.33 (2.80, 3.96)*** |
| | Substance abuse | No | | 1 | 1 | 1 |
| | | Yes | | 1.20 (0.98, 1.46) | 1.21 (0.99, 1.48) | 1.21 (1.00, 1.48) |
| | Wife beating attitude | No | | 1 | 1 | 1 |
| | | Yes | | 1.21 (1.01, 1.44) | 1.16 (0.96, 1.40) | 1.15 (0.96, 1.39) |
| | Partner's educational status | No education | | | | |
| | | Primary | | 1.62 (0.96, 2.74) | 1.58 (0.93, 2.66) | 1.56 (0.92, 2.63) |
| | | Secondary+ | | 1.58 (0.64, 3.92) | 1.50 (0.61, 3.71) | 1.50 (0.60, 3.70) |
| | Partner drinks alcohol | No | | 1 | 1 | 1 |
| | | Yes | | 3.03 (2.46, 3.74) | 3.00 (2.43, 3.69) | 3.00 (2.42, 3.67)*** |
| | Wealth index | Poorest | | 1.05 (0.80, 1.38) | 1.08 (0.82, 1.43) | 1.08 (0.82, 1.43) |
| | | Poorer | | 0.83 (0.63, 1.09) | 0.84 (0.64, 1.11) | 0.84 (0.64, 1.11) |
| | | Middle | | 1 | 1 | 1 |
| | | Richer | | 0.80 (0.59, 1.07) | 0.80 (0.60, 1.08) | 0.81 (0.60, 1.09) |
| | | Richest | | 0.72 (0.55, 1.03) | 0.76 (0.48, 1.06) | 0.74 (0.46, 1.05) |
| | Decision-making autonomy | No | | 1 | 1 | 1 |
| | | Yes | | 0.77 (0.64, 0.91) | 0.81 (0.67, 0.97) | 0.81 (0.68, 0.97)* |
| | Educational difference | Women higher | | 1 | 1 | 1 |
| | | Same | | 0.44 (0.25, 0.77) | 0.45 (0.26, 0.78) | 0.45 (0.26, 0.79)* |
| | | Husband higher | | 0.30 (0.11, 0.86) | 0.31 (0.11, 0.88) | 0.31 (0.11, 0.89)* |
| **Group** | **Variable** | **Class** | **Model I** | **Model II** AOR (95% CI) | **Model III** AOR (95% CI) | **Model IV** AOR (95% CI) |
| Level-II variables | Place of residence | Urban | | | 1 | – |
| | | Rural | | | 0.80 (0.54, 1.19) | – |
| | Early marriage | Low | | | 1 | – |
| | | High | | | 1.00 (0.79, 1.26) | – |
| | Female literacy | Low | | | 0.74 (0.57, 0.96) | 0.74 (0.57, 0.96)* |
| | | High | | | 1 | 1 |
| | Women's decision-making autonomy | Low | | | 1 | 1 |
| | | High | | | 0.79 (0.63, 0.99) | 0.80 (0.64, 1.01) |
| | IPV acceptability | Low | | | 1 | 1 |
| | | High | | | 1.22 (0.97, 1.54) | 1.31 (1.06, 1.62)* |

(*Continued*)

**Table 6.** (Continued)

| Level-III variables | MPI | Below national average | | | | 1 |
|---|---|---|---|---|---|---|
| | | Above national average | | | | 0.58 (0.33, 1.01) |
| | GEI | Below average | | | | 1 |
| | | Above average | | | | 0.85 (0.50, 1.44) |
| Random effects | | | Model I | Model II | Model III | Model IV |
| Community variance (SE) | | | 0.79 (0.11)* | 0.59 (0.10)* | 0.29 (0.26)* | 0.28 (0.26)* |
| Region variance (SE) | | | 0.20 (0.13)* | 0.19 (0.09)* | 0.17 (0.09)* | 0.12 (0.07)* |
| ICC in community (%) | | | 23.1 | 19.2 | 12.3 | 10.9 |
| ICC in region (%) | | | 4.6 | 4.7 | 4.6 | 3.4 |
| PCV_community (%) | | | Reference | 25.3 | 63.3 | 64.6 |
| PCV_region (%) | | | Reference | 5 | 15 | 40 |
| Test of Model fitness | | | Model I | Model II | Model III | Model IV |
| Likelihood ratio | | | -2336.84 | -2199.03 | -2191.73 | -2190.10 |
| AIC | | | 4679.68 | 4448.06 | 4447.16 | 4398.29 |

AOR = Adjusted Odds Ratio; CI = Confidence Interval; MPI = Multi-dimensional Poverty Index; GEI = Gender Empowerment Index; SE = Standard Error;

ICC = Intra-class Correlation Coefficient; PCV = Proportional Change in Variance; AIC = Akaike Information Criterion

*P-value ≤0.05

**P-value ≤0.01

***P-value ≤0.001

Model 1 is the empty model or a baseline model without any determinant variables; Model 2 is adjusted for individual- and relationship-level factors; Model 3 is adjusted for individual-, relationship-, and community-level factors; Model 4 is the final model adjusted for individual-, relationship-, community-, and societal-level factors

Holding other variables constant, women residing in communities with a low proportion of educated women had 26% lower (AOR 0.74, 95% CI: 0.57–0.96) odds of IPV compared to women residing in communities with a high proportion of educated women. In addition, women living in communities with high IPV accepting norms had 31% higher (AOR 1.31, 95% CI: 1.06–1.62) odds of IPV experience as compared to their counterparts. The remaining factors in the model were not significantly associated with IPV.

After the inclusion of individual-, relationship-, community-, and societal-level characteristics in Model IV, the variation in the odds of IPV experience between communities and societies still remained statistically significant with $\sigma^2 = 0.28$, p-value <0.001 and $\sigma^2 = 0.12$, p-value <0.001, respectively. As shown by the estimated ICC, 10.9% and 3.4% of the variability in IPV experience was attributable to differences between community and societal characteristics, respectively. The PCV indicated that specified factors at the three levels explained 64.6% and 40% of the variation in IPV experience across communities and societies, respectively.

## Discussion

In Ethiopia, about one in every three women has experienced IPV in their lifetime. This study showed that determinants of IPV operate at different levels in the society. At the individual-level, older age, early marriage, witnessing inter-parental violence during childhood, and an IPV accepting attitude were positively associated with IPV. At the relationship level, no decision-making autonomy in the household, higher educational attainment than partner and having a partner who drank alcohol were positively associated with IPV. At the community level, women's education and community acceptance of IPV as the norm increased the odds of IPV.

These findings reveals that multiple and inter-related factors have influence on IPV in Ethiopia that suggest the need to initiate combined interventions at different levels to reduce IPV in this country.

Women of higher age were more likely to report IPV. Different explanations have been suggested for this finding. One suggestion is that older women report their cumulative experience of IPV in their lifetime, that is, they have more time to potentially be exposed to IPV than younger women [2, 26]. On the other hand, older women might be more likely to report IPV because younger women in Ethiopia are often expected to be submissive, quiet, disciplined and loyal to their husbands and hence may have a lower probability of reporting IPV [39]. However, other researchers have found that the risk of experiencing IPV increased with younger age [1, 31, 40]. One possible reason for the contradictory findings could be cultural- and area-level differences between study samples because IPV reporting is highly dependent on the cultural acceptability of IPV, which varies by community and region. Finally, Ethiopian women from rural areas, in which the majority of them are uneducated, often do not know their exact age [27] and this could contribute to discrepancies as a result of measurement error.

Another factor related to IPV was early age at first marriage, which in Ethiopia is often arranged by families. Social practices of arranged marriage and/or early marriage are common in Ethiopia where the median age at first marriage for women is 17.1 years, which is 6.6 years less than the median age at first marriage for men [27]. These practices limit the education and development of women, and further increases the risk of IPV at an early age [41]. In Ethiopia which has a strong patriarchy and traditionally values early marriage, women who have internalized such social norms as a normal part of life might be at greater risk of IPV.

One of the views in IPV research is that IPV might be a learnt behaviour that is passed from generation to generation (i.e., inter-generational effect of violence) [42]. In the current study, women who witnessed inter-parental violence as a child were more likely to experience IPV during adult life. The finding may be explained by a phenomenon whereby women exposed to violence during early life develop attitudinal acceptance and normative understanding of violence [40]. Choi & Ting [43] described this as the 'submissive hypothesis' which implies that women who are submissive to male dominance in the family are more likely to experience IPV. Further analysis of this data using a chi-square test also revealed that there was significant association between witnessing of inter-parental violence and IPV accepting attitude ($p = 0.036$). Similarly, male partners exposed to violence as a child have an increased risk of being a perpetrator at a later age. In studies from Serbia [44], Vietnam [17], and Egypt [40], researchers revealed that men who witnessed IPV as a child were more likely to become perpetrators later in life.

In this study, women married to a partner who drank alcohol had increased odds of IPV. It has been suggested in previous research that this is due to the strong influence of alcohol on behaviour [1, 45]. For example, excessive alcohol intake may lead to thoughtless behaviour such as reduced judgment and impair the ability to understand community norms, thus increasing the chances of IPV [26, 46]. Extra expenditure on alcohol may also erode family income and may contribute to conflict that could further lead to IPV [40]. In this study, women's own substance abuse was not found to be significantly associated with IPV. This needs further investigation as the relationship between women's substance abuse and her experience of IPV is often complex. There are also views that women use substances in response to experiences of IPV, rather than their substance use being a reason of IPV.

Women's decision-making autonomy in a relationship was found to be a protective factor against IPV. In Ethiopia, a man has a mandate to control the family resources and make decisions [14, 47] and if women question or argue with their partner about resources, they may

encounter frequent abuse [48]. Other researchers also indicated that conflict arising from household finances were important predictors of spousal IPV [49].

The findings of this study revealed that neither women's education nor partner's education alone had a significant influence on IPV. However, this study has shown that differential education (women who had the same or lower education than their partner) was associated with decreased odds of IPV. The effect of educational differences was explained by Choi & Ting [43] with the 'compensation hypothesis' by which a man uses force against his wife to compensate for his inability to achieve masculine gender expectations. Moreover, women living in communities with a high prevalence of educated women were more likely to experience IPV. This might be explained by women's education being insufficient to counteract traditional gender roles of male superiority and control over his wife [50, 51]. In such contexts, men do not accept being dominated by their educated wife and may try to preserve their gender role as powerful by abusing his wife [15]. This is because in more culturally conservative areas, women's education, empowerment and autonomy are unable to change the rigid normative understanding of IPV [14, 15].

In this study, women living in communities with high IPV accepting norms were more likely to have increased odds of IPV. These gender norms create a hierarchy in relationships and inequalities that in turn affect behaviours [52]. Tolerant community norms regarding IPV that disregard some acts of violence, norms of male superiority, and perceiving IPV as an inevitable part of a relationship are basic factors that not only underlie the occurrence of IPV [16–19] but also allow it to persist in society [5, 18] and challenge intervention efforts [18]. These community and cultural norms range to the extent that they devalue IPV reporting and stigmatize women who report their abuse in order to preserve a moral order [20]. The community also has a role in maintaining the normalization of IPV through proverbs [53]. In Ethiopia, for example, proverbs such as 'a woman and a mule behave the way they are trained' are common [53]. If a male cries, he is considered 'girlish', which shows the community's attitude and tolerance towards girls suffering and crying as being normal and natural [54]. Moreover, traditional norms and gender roles affect women even when they leave their communities. For example, Ethiopian migrants living in Australia and Israel found significant patriarchal norms and IPV accepting norms within the country in which they were displaced [32, 55]. Therefore, contrary to the general perception, societal-level factors were not significantly associated with IPV, rather community-level disparities in terms of education, decision-making autonomy and IPV accepting norms were important to explaining the occurrence of IPV.

The findings of this study need to be interpreted in light of the following limitations. First, the cross-sectional nature of the study makes it difficult to determine cause and effect relationships. For example, women who are in a violent relationship might have less decision-making autonomy or might have more chance of substance use. However, future research will be needed to ascertain which event is the continuation of another. Second, despite the study strictly following WHO strategies for domestic violence research that helps to minimize under-reporting bias, under-reporting of IPV experiences may still occur due to fear of repercussions, stigma, and shame. Third, all the variables, including partner characteristics, were self-reported and might be subject to recall bias. Lastly, some factors of the ecological model such as factors related to social support to victims, neighbourhood environment, laws, and national policies were not assessed in this study due to these variables not being in the dataset.

## Conclusion and implications

In summary, the results show that the proportion of women who had experienced IPV were high in Ethiopia. This study reveals an important public health message that high IPV

prevalence was accountable to not only individual factors but also relationship- and community-level characteristics. As this study is based on robust statistical analysis and on the most representative national data, it has implications for policy makers and programmers. The evidence can be taken into account when designing future IPV prevention programs that aim to improve factors at different levels. The findings also suggest that interventions against IPV require multisectoral collaborations. It also needs the involvement of different stakeholders from communities as well as governmental and non-governmental organizations to end the intergenerational cyclic effect of IPV.

Future studies should focus on qualitative studies that might explore how the social processes cause and maintain IPV in communities. This is because even with the inclusion of many variables across different levels, this study indicates that variability of IPV was not adequately explained by the included community- and societal-level variables. This shows the complexity of the occurrence of IPV and that some other arcane social processes might be present.

## Acknowledgments

We are grateful to the Central Statistical Agency of Ethiopia and Measure Demographic and Health Survey program, which allowed us to access and use the data freely. We are also thankful to the women who participated in the survey and shared their IPV experiences. We thank the University of Newcastle, the Hunter Medical Research Institute, and the Research Centre for Generational Health and Ageing for creating a quality research environment for us to accomplish this work.

## Author Contributions

**Conceptualization:** Tenaw Yimer Tiruye.

**Formal analysis:** Tenaw Yimer Tiruye.

**Methodology:** Tenaw Yimer Tiruye.

**Supervision:** Melissa L. Harris, Catherine Chojenta, Elizabeth Holliday, Deborah Loxton.

**Writing – original draft:** Tenaw Yimer Tiruye.

**Writing – review & editing:** Melissa L. Harris, Catherine Chojenta, Elizabeth Holliday, Deborah Loxton.

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
