## [Decision Letter · Decision Letter 0]

22 Jan 2020

PONE-D-19-32625

Determinants of Intimate Partner Violence against Women in Ethiopia: A Multi-Level Analysis

PLOS ONE

Dear Tiruye,

Thank you for submitting your manuscript to PLOS ONE. After careful consideration, we feel that it has merit but does not fully meet PLOS ONE’s publication criteria as it currently stands. Therefore, we invite you to submit a revised version of the manuscript that addresses the points raised during the review process.

In addition to addressing the reviewer comments, please refer to the STROBE checklist and attach a copy to the next submission verifying where all required reporting elements are addressed. In addition, please ensure the manuscript and all supporting text destimed for publication (for example, the data availability statement) have been carefully copy-edited for appropriate professional writing style.

We would appreciate receiving your revised manuscript by Mar 07 2020 11:59PM. To enhance the reproducibility of your results, we recommend that if applicable you deposit your laboratory protocols in protocols.io, where a protocol can be assigned its own identifier (DOI) such that it can be cited independently in the future. For instructions see: http://journals.plos.org/plosone/s/submission-guidelines#loc-laboratory-protocols

We look forward to receiving your revised manuscript.

Kind regards,

Kristin Dunkle

Academic Editor

PLOS ONE

Journal Requirements:

3. Your ethics statement must appear in the Methods section of your manuscript. If your ethics statement is written in any section besides the Methods, please move it to the Methods section and delete it from any other section. Please also ensure that your ethics statement is included in your manuscript, as the ethics section of your online submission will not be published alongside your manuscript.

Reviewers' comments:

Reviewer's Responses to Questions

**Comments to the Author**

1. Is the manuscript technically sound, and do the data support the conclusions?

Reviewer #1: Yes

Reviewer #2: Yes

2. Has the statistical analysis been performed appropriately and rigorously? 

Reviewer #1: Yes

Reviewer #2: Yes

3. Have the authors made all data underlying the findings in their manuscript fully available?

Reviewer #1: No

Reviewer #2: No

4. Is the manuscript presented in an intelligible fashion and written in standard English?

Reviewer #1: Yes

Reviewer #2: Yes

5. Review Comments to the Author

Reviewer #1: This manuscript reports on the individual, relationship, and community level determinants of IPV experienced by women in Ethiopia. Overall the research appears to be rigorous. Analysis and statistical methods appear appropriate and the interpretation and conclusions are generally warranted.

Minor comments:

Pg. 6: The authors can present the items used to measure IPV in a table to improve the flow of the section.

Pg. 9, lines 151-55: The authors should explain the rationale for choosing specific cut-offs for recoding continuous community level variables into binary variables.

Pg. 11, lines 193-194: The authors should remove the question mark next to 'random effects'. There is a question mark next to random effects in Table 5 that should be removed as well.

Reviewer #2: Thank you for the opportunity to review “ Determinants of Intimate Partner Violence against women in Ethiopia: A Multi-level analysis” . The authors have tackled an interesting and important topic and have acknowledged the multi-faceted global problem of violence against women. Their application of the ecological model in understanding the determinants of IPV, makes it an interesting and important contribution to the field. Overall the paper is well written but I have a few points that I would like the authors to address:

1) In Table 1, instead of only listing level 1 variables, the authors should list all variables considered in the analysis and indicate at which level each variable has been used. This will make it easier for the reader to follow their description of variables used in the model and the modelling hierarchy.

2) Authors need to provide information on how the 3897 women are distributed across the 639 communities (PSUs).

3) Authors need to comment on the rationale behind having “Attitude to IPV” at level 1 and “Acceptance of IPV” at level 2.

4) Authors have indicated that individual and relationship level variables were added as level 1 because in the EDHS only 1 woman per HH was sampled. Yet when defining some level 2 variables , they are using the some of these level 1 variables to derive level 2 variables. Could they explain the rationale informing this? What is the mathematical implication of using same variables at level 1 and 2 albeit aggregate variables?

5) If level 2 and level 3 variables were aggregated at community and provincial level respectively, the summary statistics presented in the Tables 2 and 3 for Level 2 and 3 variables should reflect the sample size at that level (n=639 for level 2 and 11 for level 3)

6) Authors should comment on results in Table 3.

7) Authors should comment on how they decided on cut-offs for community level variables.

8) Association between age of women and lifetime experience of IPV is always difficult to entangle. I would suggest that the authors use the participant’s actual age (continuous var) as a covariate in the models rather than dwell on discussing this as possible determinant of IPV. The authors could look at whether they will get different outcome if they include age (continuous var).

9) In Line 319-320, is there evidence in the data that most of the uneducated women in the sample were from rural communities?

10) To support their assertions in line 332-334, did the authors look at whether there was an association between acceptance of IPV and witnessing of inter-parental violence?

11) In line 346-348, it is not possible to attribute causality with a cross-sectional study (as they have acknowledged in their limitations). Do not think the reason for non-significant relationship is due to assumed direction of effect. The association between factors is not dependent on what you define as an outcome.

12) Considering almost 60 % of the women and 50% of their partners have no education, it would be interesting to understand the distribution of the differential education (especially which women have high education than their partners).

13) The authors should also highlight the limitations of using secondary data vis a vis their ecological model. Were there other factors they could have included that were not available in the dataset?

14) Overall the discussion section is too long. They should make it more succinct.

6. PLOS authors have the option to publish the peer review history of their article (what does this mean?). If published, this will include your full peer review and any attached files.

Reviewer #1: Yes: Sangeeta Chatterji

Reviewer #2: No

---

## [Author Response · Author response to Decision Letter 0]

3 Mar 2020

Dear editor and reviewers,

We would like to thank you for your careful review of the draft of our manuscript entitled “Determinants of Intimate Partner Violence against Women in Ethiopia: A Multi-Level Analysis” and providing us with advice to improve the quality of the manuscript. The response to address the comments has been prepared in the summary table and attached in the submission system.

---

## [Decision Letter · Decision Letter 1]

10 Apr 2020

Determinants of Intimate Partner Violence against Women in Ethiopia: A Multi-Level Analysis

PONE-D-19-32625R1

Dear Dr. Tiruye,

We are pleased to inform you that your manuscript has been judged scientifically suitable for publication and will be formally accepted for publication once it complies with all outstanding technical requirements.

With kind regards,

Kristin Dunkle

Academic Editor

PLOS ONE

Additional Editor Comments (optional):

Reviewers' comments:

Reviewer's Responses to Questions

**Comments to the Author**

1. If the authors have adequately addressed your comments raised in a previous round of review and you feel that this manuscript is now acceptable for publication, you may indicate that here to bypass the “Comments to the Author” section, enter your conflict of interest statement in the “Confidential to Editor” section, and submit your "Accept" recommendation.

Reviewer #1: All comments have been addressed

Reviewer #2: All comments have been addressed

2. Is the manuscript technically sound, and do the data support the conclusions?

Reviewer #1: Yes

Reviewer #2: Yes

3. Has the statistical analysis been performed appropriately and rigorously? 

Reviewer #1: Yes

Reviewer #2: Yes

4. Have the authors made all data underlying the findings in their manuscript fully available?

Reviewer #1: No

Reviewer #2: Yes

5. Is the manuscript presented in an intelligible fashion and written in standard English?

Reviewer #1: Yes

Reviewer #2: Yes

6. Review Comments to the Author

Reviewer #1: The authors have been responsive to my comments and the revised paper is improved with welcome additions and clarifications.

Reviewer #2: The authors have addresses all the comments that were raised in the first round. I have np reservations on the publication of the paper.

7. PLOS authors have the option to publish the peer review history of their article (what does this mean?). If published, this will include your full peer review and any attached files.

Reviewer #1: No

Reviewer #2: No

---

## [Editor Report · Acceptance letter]

14 Apr 2020

PONE-D-19-32625R1 

Determinants of Intimate Partner Violence against Women in Ethiopia: A Multi-Level Analysis 

Dear Dr. Tiruye:

I am pleased to inform you that your manuscript has been deemed suitable for publication in PLOS ONE. Congratulations! Your manuscript is now with our production department. 

With kind regards,

on behalf of

Dr. Kristin Dunkle 

Academic Editor

PLOS ONE